# Fungal Keratitis in Northern Thailand: Spectrum of Agents, Risk Factors and Putative Virulence Factors

**DOI:** 10.3390/jof7060475

**Published:** 2021-06-11

**Authors:** Siriporn Chongkae, Sirida Youngchim, Joshua D. Nosanchuk, Angkana Laliam, Chulaluck Tangmonkongvoragul, Kritsada Pruksaphon

**Affiliations:** 1Department of Microbiology, Faculty of Medicine, Chiang Mai University, Chiang Mai 50200, Thailand; siripornannt4@gmail.com (S.C.); ang.angkanala@gmail.com (A.L.); kpruksaphon@gmail.com (K.P.); 2Department of Medicine (Infectious Diseases), Albert Einstein College of Medicine, Bronx, NY 10461, USA; josh.nosanchuk@einsteinmed.org; 3Department of Ophthalmology, Faculty of Medicine, Chiang Mai University, Chiang Mai 50200, Thailand; poupae025@gmail.com

**Keywords:** fungi, melanin, anti melanin monoclonal antibody, fungal keratitis

## Abstract

Fungal keratitis (FK) is a serious ocular infection that can result in various degrees of vision loss, including blindness. The aim of the study was to identify and retrospectively review all FK cases diagnosed between August 2012 and December 2020 at a tertiary care hospital in northern Thailand with a specific focus on epidemiologic features, including season, patient sex and age, the spectrum of pathogens, and presence of certain putative virulence factors. Of 1237 patients with corneal ulcers, 294 (23.8%) were confirmed by direct microscopic examination and/or fungal culture. For the positive cases, direct examinations of Calcofluor white (CW) stains and KOH mounts were found in 97.3% (286/294) and 76.5% (225/294), respectively (*p* < 0.05). Of the cases diagnosed by microscopy and culture, fungi were isolated in 152 (51.7%), with *Fusarium* spp. being the most frequently identified (*n* = 69, 45.5%) followed by dematiaceous fungi (*n* = 45, 29.6%) and *Aspergillus* spp. (*n* = 18, 11.8%). The incidence of FK was higher in the rainy season of July to October. The mean age was 54.4 ± 14.4 (SD) years, with a range of 9–88 years. Males (75.8%) were affected significantly more than females (24.2%) (*p* < 0.05). Of 294 patients, 132 (44.9%) were middle-aged adults (41–60 years) and 107 (36.4%) were older than 60 years. Trauma to the eye by soil or vegetative matter were the most common preceding factors (188/294; 64.0%). We assessed two virulence factors. First, 142 of the 152 culture-positive FK cases were due to molds, indicating that hyphal morphogenesis is extremely important in disease. We also demonstrated that fungal melanization occurs in the molds during the course of FK by applying a melanin-specific monoclonal antibody (MAb) that labeled fungal elements in corneal samples of patients, and melanin particles derived from the hyphae were also recovered after treatment of the samples with proteolytic enzymes, denaturant and hot concentrated acid. In summary, we demonstrate that northern Thailand has a high rate of FK that is influenced by season and males engaged in outside activities are at highest risk for disease. Moulds are significantly more commonly responsible for FK, in part due to their capacity to form hyphae and melanins. Future studies will examine models of fungal corneal interactions and assess additional factors of virulence, such as secreted enzymes, to more deeply decipher the pathogenesis of FK.

## 1. Introduction

Fungal keratitis (FK; mycotic keratitis, or keratomycosis) is a potentially sight-threatening corneal infection caused by a wide variety of filamentous fungi and yeasts [1,2,3]. The fungi invade the corneal stroma causing inflammation leading to epithelial defects and, finally, destruction of these structures, which may result in partial or complete visual loss [2]. The incidence of FK is increasing, particularly in tropical and subtropical areas [3,4,5,6]. Notably, during the last four decades, nearly half of the world’s FK cases have been reported in Asia [6,7]. The majority of FK cases are due to globally ubiquitous saprophytes and their incidence for this disease varies in different geographical locations [8].

To date, over 100 species of fungi, including filamentous fungi, yeasts and dimorphic fungi, have been identified as the aetiological agents of FK [3]. Filamentous fungi are the most common isolates in tropical and subtropical countries, which is in contrast to the predominant yeast isolates in temperate climates, European countries and USA [9,10]. Thus, a geographical difference has been shown between the prevalences of different causes of FK.

Furthermore, filamentous fungi are the major causes of post-traumatic infections, which is consistent with the variety of ecological niches associated with these species [11,12]. Among filamentous fungi, the most common organisms responsible for keratitis are *Fusarium*, *Aspergillus*, *Curvularia* and other phaeohyphomycetes, *Scedosporium apiospermum*, and *Paecilomyces*, but many diverse other species have been reported [11,12]. These fungal pathogens produce an abundance of conidia and then disperse these into the environment, and several environmental factors such as humidity, rainfall, and wind appear to have a significant impact on the prevalence of filamentous fungal keratitis [11,12].

In order to cause FK, fungal conidia have to germinate on the corneal epithelium, and undergo morphogenesis to hyphae that invade the host tissue, colonize, resist or escape host immune responses, and damage the tissue [13]. In addition to transitioning to a hyphal form, during corneal invasion, fungal pathogens utilize a number of virulence strategies to facilitate their proliferation and combat the host defenses within the cornea. Melanins, a group of biopolymers with multifunctionality, are an important virulence factors in diverse human environmental and human pathogenic fungal species [14,15,16,17,18,19]. The diverse functions of melanin enable fungi to survive in harsh conditions both in vitro and in vivo [15,17,20]. In term of the pathogenesis of human pathogenic fungi, melanin allows fungi to evade host defense response mechanisms such as reducing the toxicity of microbicidal peptides and reactive oxygen species and inhibiting their phagocytosis and killing as well as playing essential roles in fungal cell wall mechanical strength to facilitate the penetration of host tissue [15]. Melanins also reduce the efficacy of several commonly used antifungals [21].

In the present study, we evaluated temporal trends in the causative organisms of FK presenting to Maharaj Nakorn Chiang Mai Hospital, a tertiary referral center in northern Thailand, between 2012 to 2020; we also assessed for any changes in the spectrum of aetiological agents, risk factors, seasonal variation, laboratory findings and putative virulence factors of FK.

## 2. Materials and Methods

### 2.1. Fungal Morphology and Molecular Identification

A retrospective medical records review of all patients with corneal ulcers who underwent corneal scraping or corneal biopsy for the diagnosis of fungal keratitis at the Maharaj Nakorn Chiang Mai Hospital, Chiang Mai, Thailand between August 2012 and December 2020 was performed. This study was approved by the Research Ethics Committee of Chiang Mai University.

The microbiological profile of samples in the form of corneal scrapings, corneal tissue or corneal exudates were visualized as wet mount preparations using 10% potassium hydroxide (KOH) and Calcofluor white (CW) and the samples were also processed for culture. Notably, confirmation by microscopic examination and culture of the clinical samples remain the gold standard for laboratory diagnosis [22]. For culture, each sample was inoculated in the form of ‘C’ streaks on two Sabouraud’s dextrose agar (SDA) plate with chloramphenicol and incubated at 25 °C and 37 °C, respectively, and examined daily for any growth till 14 days. Any fungal growth at the point of inoculation was subcultured, and identification was performed by phenotypic, macroscopic and microscopic morphology. In addition, any fungus grown on the primary isolation medium was subcultured onto potato dextrose agar (PDA) and incubated for a period of 10 days to facilitate conidiation. Following complete growth of the fungal isolate on PDA, identification was carried out based on its macroscopic and microscopic features. The isolates that could not be identified by phenotypic methods were identified by sequencing of internal transcribed spacer (ITS) region of ribosomal DNA. Briefly, total nucleic acids were extracted by using TRIzol according to the manufacturer’s instructions (25:24:1 *v/v*, USB Corporation, Cincinnati, OH, USA). Nuclear ribosomal DNA ITS1 and ITS2 and 5.8S rDNA were amplified using ITS primers, ITS-1 (5′-TCCGTAGGTGAACCTGCGG-3′) and ITS-4 (5′-TCCTCCGCTTATTGATATGC-3′) [23]. For dematiaceous fungi, primers NL-1 (5′-GCATATCAATAAGCGGAGGAAAAG-3′) and NL-4 (5′-GGTCCGTGTTTCAAGACG-3′) were used for the amplification of the D1/D2 domains of the large subunit ribosomal DNA (LSUrDNA) [24]. The PCR products were purified with a GeneJET gel extraction kit (Thermo Scientific, Vilnius, Lithuania) and used for a sequencing reaction at a ATCG Company Limited (Thailand Science Park (TSP), Pathumthani, Thailand). Sequence analysis was performed with a Big Dye terminator Cycle Sequencing Ready Reaction kit for both strands, and the sequences were aligned with the MT Navigator software (Applied Biosystems, Foster City, CA, USA). The labelled DNA was subjected to sequencing with an ABI3730XL sequencer (Applied Biosystems). The sequence data were subjected to a BLAST search of the ITS sequence data in Genebank at the NCBI website (http://www.ncbi.nlm.nih.gov/BLAST).

In this study, a diagnosis of fungal keratitis was made according to the following criteria: (1)corneal scrapings revealed fungal elements in smears;(2)a fungus grew in more than one medium in the absence of fungal elements in smears;(3)a fungus grew on a single medium in the presence of fungal elements in smears;(4)growth of a fungus appeared at the inoculated site on culture medium.

### 2.2. Immunofluorescence Analysis of Melanin Expression in Mycotic Keratitis

To investigate melanin production, we followed a validated protocol [25]. In brief, cornea scrapings from patients with FK were obtained from the Ophthalmological Clinic, Maharaj Nakorn Chiang Mai Hospital, Chiang Mai University. Direct examination of cornea scrapings were performed by Calcofluor white (CW) staining (Sigma, St. Louis, MO, USA) and samples were concurrently cultured on Saboraud dextrose agar (SDA; Difco^TM^, Sparks, MD, USA) as described above. To examine melanization, the samples were digested with 1.0 mg proteinase K/mL (Roche, Basel, Switzerland) at 37 °C for 2 h, and washed three times with PBS. The samples were then incubated for 1.5 h at 37 °C with 10 mg/mL of a monoclonal antibody (MAb) to melanin, MAb 8D6, which was generated against melanin extracted from *A*. *fumigatus* [26]. After washing, the samples were probed with a 1:500 dilution of Alexa Fluor-488-conjugated goat anti-mouse IgM antibody (Molecular Probes, Eugene, OR, USA) for 1.5 h at 37 °C. For a negative control, conjugated goat anti-mouse IgM without primary MAb was applied. In addition, *Cryptococcus neoformans* strain H99 grown in defined minimal medium (15.0 mM glucose, 10.0 mM MgSO_4_, 29.4 mM KH_2_PO_4_, 13.0 mM glycine, and 3.0 μM vitamin B_1_ [pH 5.5]) with or without 1.0 mM L-DOPA (Sigma) for 7 days at 30 °C was used as positive and negative controls, respectively, as described [27]. Images were captured with a DS Fi1 camera (Nikon, Tokyo, Japan). In addition, slides of positive cultures of fungi were prepared as described [28]. The samples were stained with anti-melanin MAb 8D6 as detailed above.

### 2.3. Isolation of Melanin Particles from Cornea Tissues and Identification via IF Reactivity with Anti-Melanin MAb 8D6

To further confirm in vivo formation of fungal melanin in FK, the corneal materials from the patients with culture confirmed fungal infections were mounted on glass slides and subjected to a melanin extraction protocol [26]. Briefly, the corneal samples were digested with Novozyme (Sigma, St. Louis, MO, USA) (a cell-wall-lysing enzyme from *Trichoderma harzianum*), guanidine thiocyanate (denaturant), and proteinase K (Roche, Basel, Switzerland). After washing, the samples were boiled in acid then extensively washed with distilled water. The samples were then subjected to anti-melanin MAb 8D6 labelling as detailed in the in vitro work above. For a negative control, conjugated goat anti-mouse IgM without primary MAb was used. We also processed samples without fungal elements by CW or KOH, but no materials were recovered at the end of the protocol, indicating an absence of melanin.

## 3. Results

### 3.1. Epidemiological Features

Of the 294 patients with fungal keratitis, 223 (75.8%) were male, and 71 (24.2%) were female. Thus, males were significantly more affected than females (*p* < 0.005) with ratio 3:1. The mean age was 54.6 ± 14.4 years, with a range of 9 to 88 years. The age group of 51–60 years was the most frequent (*n* = 82, 27.9%), followed by those 61 to 70 years (*n* = 74, 25.2%), and 41 to 50 years (*n* = 50, 17.0%). The age and sex distribution are described in Table 1. Figure 1 shows the seasonal variation in the prevalence of FK evaluated over the period of 9 years. Our data revealed that in northern Thailand, FK occurred more often during the rainy season (July to October; 59.5% (175/294)) than in rice planning season (May to June; 14.3%) or harvesting season (November to December; 13.3%).

Predisposing risk factors were recorded in 188/294 cases (64.0%) with soil and dust 125/188 (42.5%) being the major traumatic agents involved. Plants and vegetative matter were the risk factor in 52/188 (17.7%) of the cases. Only 3.7% of FK cases occurred in individuals using contact lenses. Four cases (2.1%) occurred in individuals applying herbal medicine to their eyes. An associated risk was absent in the remaining (106/294) patients.

### 3.2. Laboratory Findings

This study included 1237 subjects with corneal ulcers, based on clinical suspicion, of whom 294 cases were confirmed as FK in the laboratory. Fungal elements of corneal scrapings were visualized by staining with CW and KOH. Among the 294 cases, the positive results of CW (97.3%; 286/294) were significantly higher than KOH staining (76.5%; 227/294) (*p <* 0.05), whereas the remaining 2.7% (8/294) cases were positive by culture only. Only 49.0% (144/294) of positive cases by direct examination were confirmed by fungal growth in culture (Figure 2). In addition, the direct examination of corneal materials by CW and KOH staining was significantly more sensitive in clinical specimens of FK when compared to culture alone (*p* < 0.05).

For the 152 fungal isolates, (the 144 with both culture and scraping visualization positive and the 8 by culture only), morphological identification was successful in 12 genera of *Fusarium*, *Aspergillus*, *Paecilomyces, Acremonium*, *Cladosporium*, *Alternaria*, *Curvularia*, *Fonsecaea*, *Phialophora*, *Pythium, Candida* and *Rhodotorula*. An additional eight genera including *Exserohilum*, *Basidiobollus*, *Bipolaris*, *Lasiodiplodia*, *Colletotrichum*, *Exophiala*, *Graphium* and *Phaeoacremonium* were identified by sequencing. *A. flavus* and *C. parapsilosis* were confirmed at the species level by molecular identification. *Fusarium* spp. (45.4%) were the most common fungal species isolated followed by dematiaceous fungi (29.6%) and *Aspergillus* spp. (11.8%). Yeasts, including *Candida* spp. and *Rhodotorula* spp., were only found in 6.6% cases (Table 2). In eight patients, we detected both filamentous fungi and yeasts; the four filamentous fungi were a *Fusarium* spp., *Paecillomyces* sp, *Cladosporium* spp., and an unidentified dematiaceous fungus whereas the remaining yeast isolates were 3 *Rhodotorula* spp. and 1 *C. parapsilosis*.

Notably, the number of FK cases increased over the first eight-years of retrospective study at the Maharaj Nakorn Chiang Mai Hospital with a slight decline in 2019 (Figure 3). In 2020, the COVID-19 pandemic complicated further study of FK patients as the hospital limited patients with other diseases to focus on SARS-CoV-2, which led us to terminate our review with patients through 2019.

### 3.3. Melanin Expression in Mycotic Keratitis by Anti-Melanin MAb 8D6

In order to detect fungal melanization in FK, we used the anti-melanin MAb 8D6 to assess reactivity to fungal elements within corneal materials. In hyaline fungi, 16 isolates of *Fusarium* spp., 12 isolates of *Aspergillus* spp. and one isolate of *Paecilomyces* were positive for melanin staining. Melanin staining was also positive in dermatiaceous fungi including eight isolates of unidentified dematiaceous fungi, two isolates of *Cladosporium* spp., one isolate of *E*. *rostratum*, *B*. *hawaiiensis*, *L*. *theobromae*, *Alternaria* sp., *C*. *gloeosporioides*, *Curvularia* sp., *F*. *pedrosoi*, *G*. *kuroshium* and *P*. *verrucosa*. MAb 8D6 efficiently labeled melanin in the cell wall of hyphae in the FK samples (Figure 4A,B and Figure 5A,B). Our melanin extraction treatments of the corneal samples containing fungal elements resulted in dark particles identical in size and shape to hyphae that were also positive with anti-melanin MAb by IF (Figure 4C,D and Figure 5C,D). Melanin production was confirmed in the filamentous fungal cultures (Figure 4E,F and Figure 5E,F). 

Figure 4 and Figure 5 are representative examples of the dematiaceous (*F**. pedrosoi*) and hyaline (*A*. *flavus*) fungi identified. We tested 48 representative molds and six yeasts for the presence of melanin. All molds were labeled by the melanin-binding MAb. No labeling of yeast cells was noted.

We also studied exudates from patients without positive fungal cultures. Similar to the staining by CM and/or KOH, the melanin-binding MAb efficiently labeled the cell wall of the fungal elements, demonstrating melanin synthesis in these fungal negative culture FK cases (Figure 6).

## 4. Discussion

We reviewed the epidemiological features of 294 patients with mycotic keratitis seen over a period of eight years at a tertiary hospital in northern Thailand. Our studies confirmed that cornea trauma was the most common risk factor (83.4%), especially injury from contaminated soil (50.5%) and plant matter (32.9%), which is similar to FK described in other tropical climates, such as India [6,29], the Philippines and China [7,30]. Also, the highest incidence, 45.2%, of FK occurred in the rainy season (July to October) in Thailand, where many environmental factors such as humidity, temperature, rainfall and wind extensively influence the occurrence of filamentous FK resulting more contact to those fungi that thrive in this environment [9]. Recent reports have indicated that the incidence of FK was 32.8% in 8 Asian countries including India, China, Japan, South Korea, Taiwan, Thailand, the Philippines, and Singapore, and only the incidence was only 9.1% in southern Thailand during the 12–18 month study period [7]. In line with prior studies both in the southern part of Thailand and other countries, males were significantly more frequently affected than females (a ratio of 3:1) [31,32]. In these regions, men are more frequently involved in outside work, including construction and agriculture, thereby enhancing their risks for infections from environmental pathogens. Similar to other reports in tropical countries, our study found that the patients 46 years of age and older (76.4%), especially middle-aged adults (46–60 years), most frequently developed FK [3,7].

Our results indicate that exposure to soil, especially during the rainy season, was the critical risk factor for acquiring fungal pathogens in the tropical areas of Thailand. Based on the widespread distribution of fungal pathogens in the environment, they are recognized as causative agents of significant diseases in both major crops and immunocompromised humans [33]. *Fusarium* spp. and *Aspergillus* spp. have previously been the most commonly reported fungal pathogens isolated from patients with FK in Asia [7,34]. As reported in other tropical countries [35], our study found that *Fusarium* spp., dematiaceous fungi and *Aspergillus* spp. were the predominant causative agents responsible for FK in northern Thailand. Prior work in northern India also reported 21.6% dematiaceous melanized fungi and 47.6% *Aspergillus* spp. [6] and a study in the western region of Nepal [36] found that 41.8% of 686 cases were due to dematiaceous fungi including unidentified dematiaceous fungi (22.0%), *Curvularia* sp. (17.7%) and *Bipolaris* sp. (2.1%). Although there is regional variation in the spectrum of agents of FK, melanized fungi appear to be increased in tropical and semitropical areas.

Melanins have been linked to fungal pathogenesis of plants, particularly during fungal invasion and penetration into the plant for successful turgor development [37]. We previously reported melanogenesis of filamentous fungi by demonstrating the presence of melanins in external layers of hyphal cell walls in both dematiaceous and hyaline fungi during cornea infection [16]. In FK, melanin may protect fungal cells from host defense mechanisms during initial stages of infection. In fact, antimicrobial peptides (AMPs) such as β-defensins and cathelicidins produced by epithelial cells at ocular surface are the first line of defense against invading pathogens [38]. However, fungal melanin inactivates antimicrobial peptides as well as neutralizes oxidative stresses to attenuate fungicidal activity due to inflammatory responses [39].

Overall, we report that FK occurs with a variable but relatively high frequency in northern Thailand. There is a close association for disease risk with season and outdoor activity, particularly in middle-aged males. Saprophytic molds predominate as the cause of disease and melanin formation appears to significantly contribute to pathogenesis.

## Figures and Tables

**Figure 1 jof-07-00475-f001:**
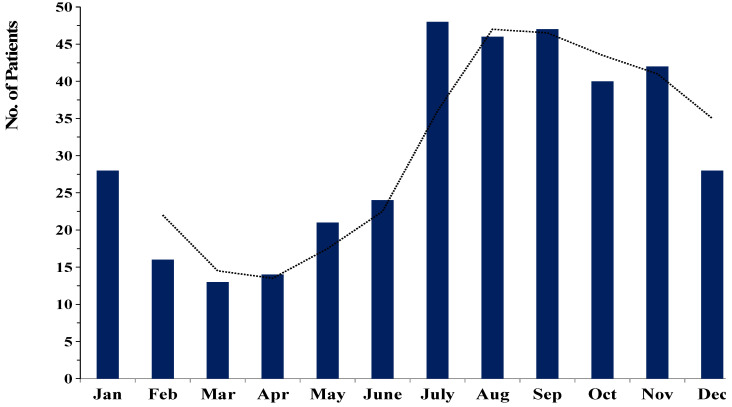
Seasonal trends in the occurrence of fungal keratitis cases between 2012 and 2020. (The dotted line was represented the trend of the prevalence of FK in each month.).

**Figure 2 jof-07-00475-f002:**
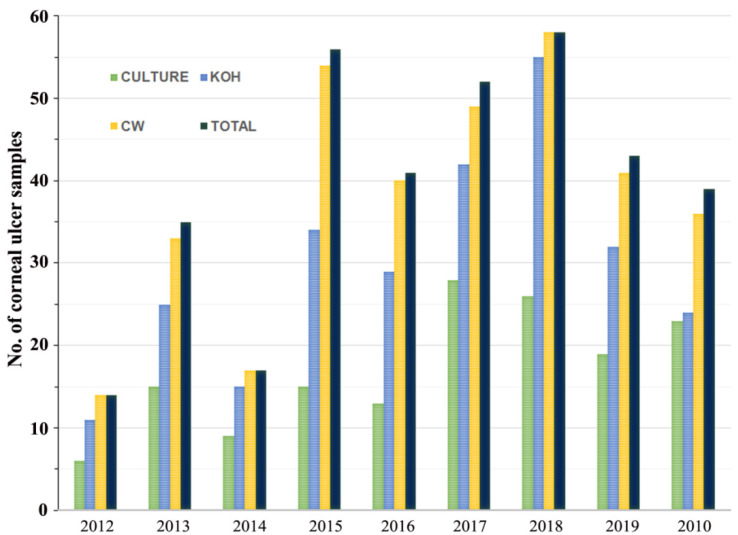
Comparative study of positive results for Calcofluor white (CW), KOH wet mount, and culture-proven disease relative to total cases of fungal keratitis identified.

**Figure 3 jof-07-00475-f003:**
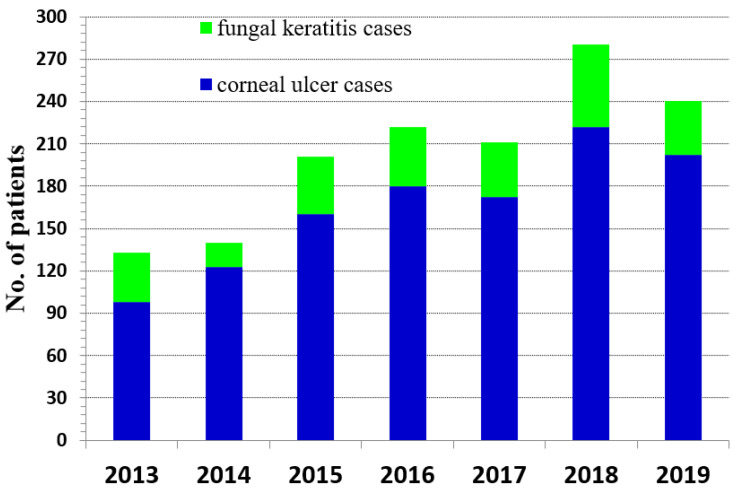
Number of patients with confirmed fungal keratitis compared to patients with other causes of corneal ulcers from 2013 to 2019.

**Figure 4 jof-07-00475-f004:**
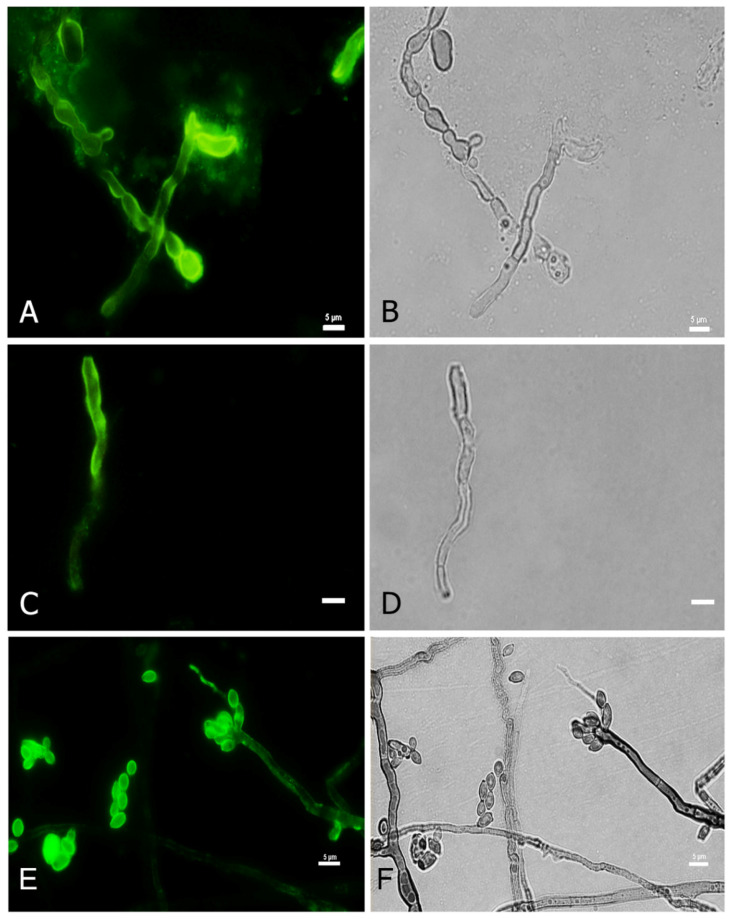
Melanin-specific MAb 8D6 labels the cell walls of septate hyphae of fungal keratitis caused by *F. pedrosoi*. Corresponding immunofluorescence and bright-field microscopy images demonstrating the labelling with MAb 8D6 of the cell walls of hyphae in a corneal sample of FK (**A**,**B**), melanin extracted particles of hyphae recovered from the patient’s corneal material (**C**,**D**) and melanin expression in the associated *F*. *pedrosoi* culture (**E**,**F**). Bars, 5 μm.

**Figure 5 jof-07-00475-f005:**
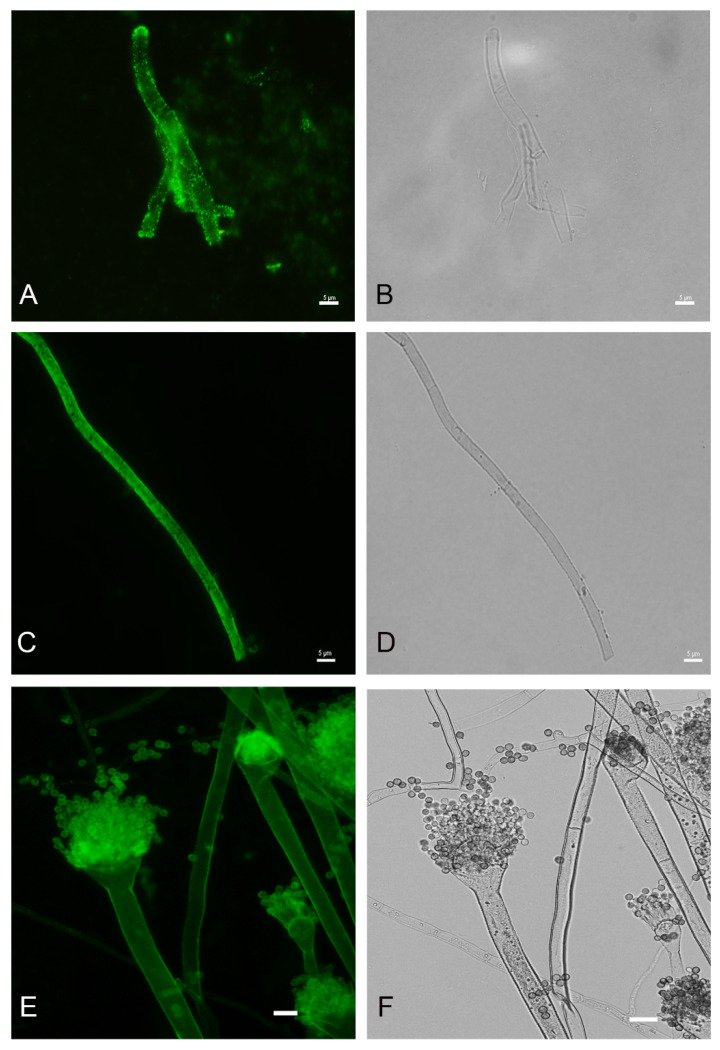
Melanin-specific MAb 8D6 labels the cell walls of septate hyphae of fungal keratitis caused by *A. flavus*. Corresponding immunofluorescence and bright-field microscopy images demonstrating the labelling with MAb 8D6 of the cell walls of hyphae in a corneal sample of FK (**A**,**B**), melanin extracted particles of hyphae recovered from the patient’s corneal material (**C**,**D**) and melanin expression in the associated *A. flavus* culture (**E**,**F**). Bars, 5 μm.

**Figure 6 jof-07-00475-f006:**
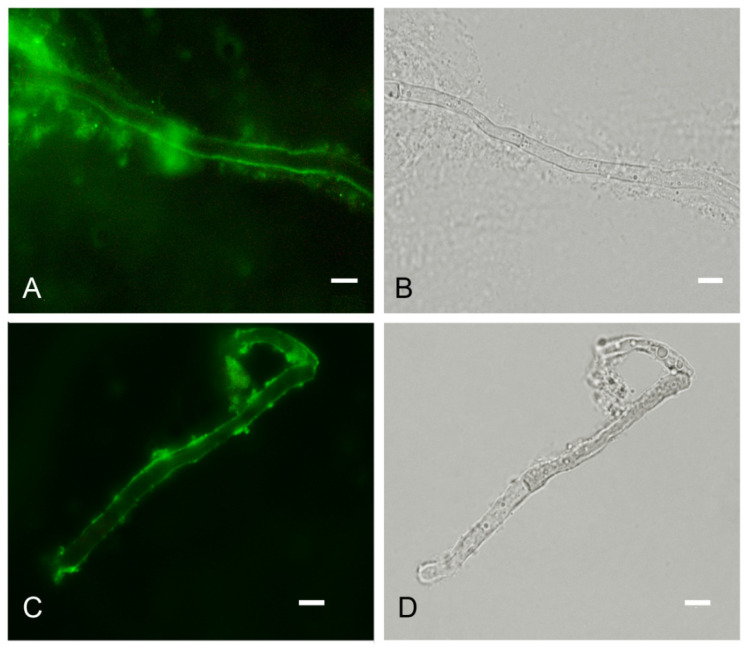
Melanin-specific MAb 8D6 labels the cell walls of septate hyphae in corneal exudate obtained from a patient with culture-negative fungal keratitis. Corresponding immunofluorescence (**A**) and bright-field (**B**) images demonstrating the labelling of the cell walls of septate hyphae by the MAb in cornea scraping. Corresponding immunofluorescence (**C**) and bright-field (**D**) images showing melanin particles recovered from corneal sample labeled with MAb 8D6. Bars, 5 μm.

**Table 1 jof-07-00475-t001:** Age and sex distribution of mycotic keratitis patients.

Age (Years)	Male (%)	Female (%)	Total (%)
** 0–10 **	2 (0.9%)	0	2 (0.7%)
** 11–20 **	2 (0.9%)	0	2 (0.7%)
** 21–30 **	12 (5.4%)	2 (2.8%)	14 (4.8%)
** 31–40 **	33 (14.8%)	4 (5.6%)	37 (12.6%)
** 41–50 **	35 (15.7%)	15 (21.1%)	50 (17.0%)
** 51–60 **	62 (27.8%)	20 (28.2%)	82 (27.9%)
** 61–70 **	51 (22.9%)	23 (32.4%)	74 (25.2%)
** >70 **	26 (11.6%)	7 (9.9%)	33 (11.2%)
** Total **	** 223 ** ** (75.8%) **	** 71 ** ** (24.2%) **	294 **(100%)**

**Table 2 jof-07-00475-t002:** Spectrum of fungal agents isolated during the study period of fungal keratitis.

Type of Fungi	No. of Isolates	%
**Filamentous hyaline fungi**		
- *Fusarium* spp.	69	45.4
- *Aspergillus fumigatus*	8	5.2
- *Aspergillus flavus*	6	3.9
- *Aspergillus *spp.	4	2.6
- *Paecilomyces* sp.	1	0.7
- *Acremonium* sp.	1	0.7
- Hyaline fungi nonsporulated	1	0.7
**Dematiaceous fungi**		
- Dematiaceous fungi nonsporulated	24	15.7
- *Cladosporium* spp.	4	2.6
- *Exserohilum rostratum*	3	2.0
- *Basidiobolus ranarum*	2	1.3
- *Bipolaris hawaiiensis*	2	1.3
- *Lasiodiplodia theobromae*	2	1.3
- *Alternaria *sp.	1	0.7
- *Colletotrichum gloeosporioides*	1	0.7
- *Curvularia* sp.	1	0.7
- *Fonsecaea pedrosoi*	1	0.7
- *Exophiala jeanselmei*	1	0.7
- *Graphium kuroshium*	1	0.7
- *Phaeoacremonium parasiticum*	1	0.7
- *Phialophora verrucosa*	1	0.7
**Oomycetes**		
- *Pythium insidiosum*	7	4.6
**Yeasts**		
- *Candida albicans*	3	2.0
- *Candida parapsilosis*	1	0.7
- *Candida *spp.	2	1.3
- *Rhodotorula* spp.	4	2.6
**Total**	**152**	**100.0**

## Data Availability

All data related to this manuscript is incorporated in the manuscript only.

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
