# Peer review of "Fungal Keratitis in Northern Thailand: Spectrum of Agents, Risk Factors and Putative Virulence Factors"

_jof, 2021, doi:10.3390/jof7060475_

Round 1
Reviewer 1 Report
The authors present an interesting review of FK cases in their clinics in Thailand. It provides useful information on the risk factors, microbiology and seasonality of this potentially blinding infections. The writing is good. My suggestions are as follows. My apologies if the answers were present in the manuscript and I missed them.
- Definition of suspected cases would also be of help- were there specific clinical findings which triggered suspicion.
- Were there positive and negative controls for melanin staining. For instance, did they had positive and negative lab strains they tested in parallel when they were doing the clinical testing. It was mentioned that the candida cases were negative
- More descriptive legend for Fig 2 and explanation of numbers. It seems that less than 50% of cases were culture positive but on line 164 it is stated that “50.4% (144/286) of the positive cases by direct examination were confirmed by culture”. Later in line 168 it say that there were “152 cases of culture proven FK”. So I am unclear about the numbers.
- As written, it is unclear to me how many cases were tested for melanin and if any of the filamentous fungi were negative. If all the filamentous cases were positive it is hard to conclude that this was a key virulence factor rather than just a marker for filamentous fungi. If this is the case, then I don't think that it is fair to comment on its contribution to virulence, I would just say that it is a feature of the filamentous fungi in this series.
Author Response
Reviewer 1#
- Definition of suspected cases would also be of help- were there specific clinical findings which triggered suspicion.
Response: For clarity, all “clinical suspect cases” were changed to “patients with corneal ulcers”. (line 30, and Figure 3)
- Were there positive and negative controls for melanin staining. For instance, did they had positive and negative lab strains they tested in parallel when they were doing the clinical testing. It was mentioned that the candida cases were negative
Response: Thank you very much for your suggestion, and we apologize for not previously including this information. When staining fungal melanin, positive and negative controls were included in every experiment. We added the information regarding positive and negative controls in the Methods. (line 143-146 ).
“In addition, Cryptococcus neoformans strain H99 grown in defined minimal medium (15.0 mM glucose, 10.0 mM MgSO4, 29.4 mM KH2PO4, 13.0 mM glycine, and 3.0 μM vitamin B1 [pH 5.5]) with or without 1.0 mM L-DOPA (Sigma) for 7 days at 30°C were used as positive and negative controls, respectively, as described”
- More descriptive legend for Fig 2 and explanation of numbers. It seems that less than 50% of cases were culture positive but on line 164 it is stated that “50.4% (144/286) of the positive cases by direct examination were confirmed by culture”. Later in line 168 it say that there were “152 cases of culture proven FK”. So I am unclear about the numbers.
Response: Thank you for pointing out that we didn’t present this information as clearly as we thought. The 152 cases of culture proven FK were from 144 patients with samples that were positive upon direct examination and culture positive as well as 8 cases of FK that were negative by direct examination but culture positive. We have revised the text appropriately to make this very clear.
- As written, it is unclear to me how many cases were tested for melanin and if any of the filamentous fungi were negative. If all the filamentous cases were positive it is hard to conclude that this was a key virulence factor rather than just a marker for filamentous fungi. If this is the case, then I don't think that it is fair to comment on its contribution to virulence, I would just say that it is a feature of the filamentous fungi in this series.
Response: This is a fair statement, though melanin is well demonstrated to be an essential factor of virulence in diverse fungi. We are now more clearly indicating that this virulence factor is associated with the disease. Future studies in model systems where we can block melanin will be developed to elucidate the role of this enigmatic pigment in disease pathogenesis. We tested 48 representative molds and 6 yeasts, with all molds staining positive for melanin.
Reviewer 2 Report
An interesting article, contributing to knowledge of aetiology and epidemiology of fungal keratitis.
- There are some inconsistencies in the presentation of findings and data. In the abstract the study sample is described as '1,237 clinically suspected FK cases' (line 16) but the results state '1,237 subjects with corneal ulcers' (line 159) - clarity is required.
- It would be preferable to describe these fungi as fungal pathogens rather than saprophytes because many cause disease in living tissue not only feeding off dead organic matter.
- The use of spp is for multiple species, if there is just one organism or species sp should be used
- "First, 142 of? 152 culture-positive cases..." (line 27)
- 'Rainy season' (line 22)
- more recent reviews suggest >100 species of fungi have been reported to cause FK (line 49)
- What is meant by a corneal intervention needs to be defined (line 84/85)
- Replace 'C-shapes' with 'C-streaks' (line 92)
- 'at point of inoculation' to replace 'on the C' (line 95)
- Provide details of sequencing, which sequencer, where/by whom, kit/protocol to prepare sample, how many isolates was this needed for, etc (lines 100/101)
- Need to clarify confluent growth (4) (line 105) - it is very easy for a single colony to cover the surface and this could be construed as confluent. No mention that microscopy is gold standard.
- Clarity over predisposing risk factors - not all listed, even though trauma is number one RF worldwide, predisposing RFs do vary between countries/regions and urban/rural settings (lines 146/7)
- Fig 2 is somewhat confusing. The total +'ve tests are added together, so total number of patients misrepresented. Presumably, for some patients all 3 test were positive but the figure does not capture this. Would suggest altering the way this is presented.
- Table 2 - sp/spp corrections and Pythium insidiosum put into a separate category headed 'Oomycetes' rather than included in the dematiaceous fungal group
- Is sentence in lines 191/192 correct?
- Sections 3.1 and 3.4, are this incorrectly labelled or are 3.2 and 3.3 missing?
- 'Only 50.4% (144/286)' (line 164) does not correspond with 152 culture +'ve cases mentioned in abstract and in line 168
- dematiaceous (line 242)
Additional comments:
Although hyphal morphogenesis (line 27/28) is attributed to moulds, some Candida spp. also produce hyphae in vivo. There are other physiological factors associated with the site of infection which could be mentioned here in addition to this.
Were there any limitations / challenges encountered with the melanin studies?
Author Response
- There are some inconsistencies in the presentation of findings and data. In the abstract the study sample is described as '1,237 clinically suspected FK cases' (line 16) but the results state '1,237 subjects with corneal ulcers' (line 159) - clarity is required.
Response: Thank you for your comment. The clinically suspected FK cases have been changed to corneal ulcer cases
- It would be preferable to describe these fungi as fungal pathogens rather than saprophytes because many cause disease in living tissue not only feeding off dead organic matter.
Response: Thank you for your suggestion. The saprophytic fungi usages have been replaced with fungal pathogens.
- The use of spp is for multiple species, if there is just one organism or species sp should be used
Response: Thank you, we have corrected this mistake.
- "First, 142 of? 152 culture-positive cases..." (line 27)
Response: Thank you for pointing out an error. We have corrected the error.
- Rainy season' (line 22)
Response: It is changed to “rainy season”.
- more recent reviews suggest >100 species of fungi have been reported to cause FK (line 49)
Response: It is changed to over 100 species. (line 61)
- What is meant by a corneal intervention needs to be defined (line 84/85)
Response: Corneal intervention is changed to corneal scraping or corneal biopsy.
- Replace 'C-shapes' with 'C-streaks' (line 92)
Response: It is changed from “C-shape” to “C-streaks”
- 'at point of inoculation' to replace 'on the C' (line 95)
Response: It is changed to “at point of inoculation”
- Provide details of sequencing, which sequencer, where/by whom, kit/protocol to prepare sample, how many isolates was this needed for, etc (lines 100/101)
Response: The details of sequencing were added in section of Methods 2.1.
We add the number of isolates that identify by sequencing in 3.2 Laboratory Findings.
- Need to clarify confluent growth (4) (line 105) - it is very easy for a single colony to cover the surface and this could be construed as confluent. No mention that microscopy is gold standard.
Response: It is changed to “growth of the fungal colony appeared at the inoculated site on culture medium.”
We also add “Notably, confirmation by microscopic examination and culture of the clinical samples remain the gold standard for laboratory diagnosis” in Methods 2.1 Fungal morphology and molecular identification
- Clarity over predisposing risk factors - not all listed, even though trauma is number one RF worldwide, predisposing RFs do vary between countries/regions and urban/rural settings (lines 146/7)
Response: We have added additional information to address this question.
- Fig 2 is somewhat confusing. The total +'ve tests are added together, so total number of patients misrepresented. Presumably, for some patients all 3 test were positive but the figure does not capture this. Would suggest altering the way this is presented.
Response: Thank you for this suggestion. We have revised both Fig 2 and 3 for improved clarity of the data.
- Table 2 - sp/spp corrections and Pythium insidiosum put into a separate category headed 'Oomycetes' rather than included in the dematiaceous fungal group
Response: Pythium insidiosum is classified in Oomycetes as suggestion.
- Is sentence in lines 191/192 correct?
Response: Yes, and we have further clarified this paragraph.
- Sections 3.1 and 3.4, are this incorrectly labelled or are 3.2 and 3.3 missing?
Response: Thank you for pointing this out the error. Sections are changed to 3.1, 3.2 and 3.3.
- 'Only 50.4% (144/286)' (line 164) does not correspond with 152 culture +'ve cases mentioned in abstract and in line 168
Response: The total isolates of fungi were 152 isolates which consisted of 144 positive cases by direct examination and confirmed by fungal growth in culture. The remaining of 8 cases were from positive cultures only. 'Only 50.4% (144/286)' is changed to “ Only 49.0% (144/294)”
- dematiaceous (line 242): it is unclear what this refers to, but we have reviewed the adjacent text to this line.
Additional comments:
Although hyphal morphogenesis (line 27/28) is attributed to moulds, some Candida spp. also produce hyphae in vivo. There are other physiological factors associated with the site of infection which could be mentioned here in addition to this.
Response: In our study, only 5 cases of Candida spp. were collected and positive for CW in corneal samples. Notably, staining revealed only budding yeast cells; there were no hyphae.
From the study of Sun et al. (2007), 29 smears of Candida keratitis were investigated and found 15 positive smears of fungal elements (7 showing yeasts, 6 yeasts and pseudohyphae, and 2 pseudohyphae only).
Reference: Sun RL, Jones DB, Wilhelmus KR. Clinical characteristics and outcome of Candida keratitis Am J Ophthalmol . 2007 Jun;143(6):1043-1045. )
Were there any limitations / challenges encountered with the melanin studies?"
Response: We did not encounter issues with this approach as we have extensive experience with staining for melanin with our melanin-specific monoclonals as demonstrated by our ~30 publications https://pubmed.ncbi.nlm.nih.gov/?term=nosanchuk+and+melanin+and+antibody&sort=pubdate
Reviewer 3 Report
This paper examines the features of fungal keratitis in Thailand over a relatively long period of time. The authors show more frequent occurrence in males, predisposing factors of ocular trauma, a seasonal variation (rainy season being more common), and they report the types of fungi causing the disease. They also add in a method for examining melanin expression by the fungi during infection.
Overall The paper is well written and contains new information. The long time period allows for good data on types of fungi and seasonality of the disease.
My only major comment would be to see if the gender and predisposing factors such as trauma to the eye are related. It may be that males are more likely to suffer trauma than females?
Author Response
My only major comment would be to see if the gender and predisposing factors such as trauma to the eye are related. It may be that males are more likely to suffer trauma than females?
Response: In Thailand, the two most common occupations of patients with fungal keratitis are 1) farmers and 2) outdoor laborers (e.g. construction, landscapers, etc), and males predominate in these careers. We agree that these occupations and ongoing potential exposures are why males are more affected by FK compared to females. We have further clarified the text.